# Pharmacological Approaches to Modulate the Scarring Process after Glaucoma Surgery

**DOI:** 10.3390/ph16060898

**Published:** 2023-06-19

**Authors:** Debora Collotta, Simona Colletta, Virginia Carlucci, Claudia Fruttero, Antonio Maria Fea, Massimo Collino

**Affiliations:** 1Department of Neurosciences “Rita Levi Montalcini”, University of Turin, 10126 Turin, Italy; debora.collotta@unito.it (D.C.); simona.colletta@edu.unito.it (S.C.); massimo.collino@unito.it (M.C.); 2Chemsafe S.r.l., Via Ribes 5, 10010 Colleretto Giacosa, Italy; v.carlucci@chemsafe-consulting.com; 3Hospital Pharmacy, S. Croce e Carle Hospital Cuneo, 12100 Cuneo, Italy; cfruttero71@gmail.com; 4Struttura Complessa Oculistica, Città Della Salute e Della Scienza di Torino, Dipartimento di Scienze Chirurgiche, Università degli Studi di Torino, 10126 Torino, Italy

**Keywords:** glaucoma, fibrosis, intraocular pressure, glaucoma surgery

## Abstract

Glaucoma is an acquired optic neuropathy that results in a characteristic optic nerve head appearance and visual field loss. Reducing the IOP is the only factor that can be modified, and the progression of the disease can be managed through medication, laser treatment, or surgery. Filtering procedures are used when target pressure cannot be obtained with less invasive methods. Nevertheless, these procedures require accurate control of the fibrotic process, which can hamper filtration, thus, negatively affecting the surgical success. This review explores the available and potential pharmacological treatments that modulate the scarring process after glaucoma surgery, analyzing the most critical evidence available in the literature. The modulation of scarring is based on non-steroidal anti-inflammatory drugs (NSAIDs), mitomycin, and 5-fluorouracil. In the long term, the failure rate of filtering surgery is mainly due to the limitations of the current strategies caused by the complexity of the fibrotic process and the pharmacological and toxicological aspects of the drugs that are currently in use. Considering these limitations, new potential treatments were investigated. This review suggests that a better approach to tackle the fibrotic process may be to hit multiple targets, thus increasing the inhibitory potential against excessive scarring following surgery.

## 1. Introduction

Glaucoma is a chronic progressive disease that requires the continuous long-term engagement of the patient. Although several risk factors such as age, positive family history, African American descent, myopia, diabetes, and hypertension can contribute to the development and progression of glaucoma, intraocular pressure (IOP) is the only modifiable factor with a significant impact on the progression of the disease. Increased IOP in glaucoma is mainly caused by a reduced aqueous humor outflow facility rather than an increased production rate of aqueous humor. Glaucoma results in a gradual and irreversible loss of peripheral vision; patients often underestimate this symptom that, without proper treatments, can lead to blindness. In Europe, the prevalence of this disease is 2.93% in the population aged between 40 and 80 years, but increases to 10% in the population over 90 years of age [1]. The IOP has an average value between 15 and 16 mmHg (with a standard deviation of 3 mmHg). A value of 21 mmHg (>95% CI of the mean IOP of the healthy population) is set as the limit of normal IOP, which is mainly based on statistical evidence rather than clinical considerations.

## 2. Therapeutic Approaches for Glaucoma

The first line of treatment for glaucoma is pharmacological therapy, but surgery may be required if the IOP remains elevated or if the damage progresses. The pharmacological classes in use include the following: beta-blockers, alpha-2-agonists, prostaglandin (PG) analogues, carbonic anhydrase inhibitors, and Rho-associated kinase inhibitors.

Laser trabeculoplasty is widely used to treat open-angle glaucoma, a type of glaucoma in which the angle between the iris and cornea is open, but the outflow through the trabecular meshwork that is responsible for draining aqueous humor becomes less efficient over time. It can be either an alternative to pharmacological therapy or used when topical therapy fails to maintain the IOP within the target range, before considering surgical options. Laser therapy has a good safety profile, but its effect fades over 2 to 5 years in a relatively high percentage of patients that are already involved in pharmacological therapy. With selective laser trabeculoplasty (SLT), using a pulsed ND-Yag laser, the radiation energy is selectively absorbed by a pigmented cell population, thus causing less collateral damage when compared to previous methods and allowing for retreatment [2].

Surgical therapy is often considered when laser or pharmacological treatment fails to achieve the target IOP. The IOP reduction can be obtained by increasing the aqueous humor outflow or reducing the aqueous humor production. The latter is usually considered for patients with special forms of glaucoma, with a high IOP and poor visual acuity, or patients who cannot undergo other surgeries because of poor general conditions or the impossibility of a proper follow-up. Surgeries that improve the aqueous humor drainage can be classified based on which aqueous structure the humor is directed toward. Some procedures aim to open up the physiologic outflow pathways, while others shunt the aqueous humor toward the suprachoroidal or subconjunctival space. The choice of the surgical approach depends on the specific case and the severity of the glaucoma. Presently, the techniques that shunt the aqueous in the subconjunctival space are the procedure of choice, but their efficacy, especially in the long term, depends on the control of the scarring process.

### 2.1. The Scarring Process

When a surgical procedure produces an acute wound, the healing process can result in excessive scarring, determining a progressive obstruction of the surgical outflow.

The healing after glaucoma surgery can be divided into different and partially intertwined phases. The first phase involves replacing damaged cells with cells of the same type, and it does not lead to the formation of scars. The second involves replacing damaged cells with connective tissue, resulting in scar formation. The repair process occurs in all organs and body districts after injuries. In the process, various cellular and extracellular signaling pathways are involved in a linked and coordinated manner. This process is divided into the following four phases: hemostasis, inflammatory phase, proliferative phase, and tissue remodeling.

### 2.2. The Wound Healing Process after Glaucoma Surgery

The healing response and the subsequent fibrosis determine the long-term IOP. Although new and less invasive surgical techniques were introduced, the scarring process is still the major limiting factor for all subconjunctival glaucoma procedures.

The transforming growth factor-beta (TGF-β), vascular endothelial growth factor (VEGF), platelet-delivered growth factor (PDGF), fibroblast growth factor (FGF), and connective tissue growth factor (CTGF) are examples of the pleiotropic factors that are involved in conjunctival fibrosis. These factors also help to adequately regulate the healing process [3].

Modern therapies target inflammation and proliferation. The mechanism of hemostasis can be divided into various steps including blood vessel constriction, the formation of a temporary “platelet plug”, the activation of the coagulation cascade, and the formation of a “fibrin plug” or the final clot. The inflammatory phase, which is activated during the coagulation phase, is characterized first by the recruitment of neutrophils and later by the presence of monocytes [4].

Considering the proliferative phase, in the next 5–14 days, the focal point of the healing process is the recovery of the wound surface [5]. During the remodeling phase, the granulation tissue matures into a scar, increasing the tissue’s tensile strength (Figure 1).

## 3. Pharmacological Approaches Targeting the Inflammatory Response after Glaucoma Surgery

Inflammation is a limiting factor for surgical success, and the outcomes of filtration surgery and the ocular surface condition are closely related. Some IOP-lowering compounds and preservatives were shown to negatively impact the health of the conjunctival tissue and are believed to adversely affect the final surgical results. Thus, various techniques were proposed to “prepare” the conjunctival tissue for surgery, from replacing topical therapy with systemic therapy, to discontinuing the most inflammatory compounds (prostaglandin analogs and alpha agonists), to switching to unpreserved medications and using steroids or non-steroidal drugs. Nevertheless, adopting these measures is not always feasible due to practical problems and the lack of precise guidelines [7]. Table 1 and Figure 2 summarize the mechanism and characteristics of the drugs that are currently used and under investigation for controlling the scarring process.

### Topical Steroids and Non-Steroidal Anti-Inflammatory Drugs (NSAIDs)

The use of steroid drugs in preparation for surgery and the post-operative period (especially topical steroids, such as prednisone acetate 1%) is preeminent. Steroids can have many positive effects as anti-inflammatory agents, such as reducing vascular permeability, swelling, and phagocytosis; releasing growth factors; and inhibiting fibroblasts, cytokines, and macrophages, thus reducing scarring. Unfortunately, steroids can produce side effects because they cause immunosuppression, hasten the formation of posterior subcapsular cataract formation, and may cause elevated intraocular pressure. Individuals who develop an increased IOP following steroid use are “steroids responders”. The possibility of a high response to steroids includes patients with a family history of primary open-angle glaucoma, diabetes mellitus, high myopia, and connective tissue disorders such as rheumatoid arthritis. The elderly population is more susceptible, as are children less than six years of age. Nevertheless, it should be observed that the IOP rise determined by steroids in the presence of a filtering surgery is usually minimal or absent [6,8].

Several studies investigated methods of administration, other than eye drops, to raise the steroid bioavailability and effectiveness in dampening the inflammatory response. Synthetic macromolecules such as glucosamine and cyclodextrins were developed to enhance steroid delivery in experimental studies. The encapsulation of drugs in colloidal liposomes and nanoparticles increases permeation across cell membranes and prevents enzymatic degradation [9]. A dexamethasone ocular implant, which is traditionally used to treat retinal edema, was tested as an anti-scarring agent in a pilot study consisting of three female patients who underwent trabeculectomy. During the surgery, a subconjunctival 0.7 mg sustained-release dexamethasone implant was positioned against the sclera posterior to the surgical flap; after seven days, one month, and two months, the implant seemed to fragment, reducing its dimensions, and was visually undetectable after two months. In these eyes, conjunctival scarring at the site of the filtering bleb was absent. No ocular adverse events or post-operative complications were observed [10] due to the poly-2-hydroxyethyl methacrylate hydrogels, which provide the correct amount of drug release [11].

On the other hand, NSAIDs can be associated with side effects such as bleb leakage, which is why their use is still disputable. Prostaglandin E2 (PGE2) breaks the blood–aqueous barrier and raises the IOP (but prolonged exposure to PGE2 leads to a reduction in the IOP by increasing uveoscleral outflow) [12], whereas prostaglandin F2 (PGF2) typically lowers the IOP by significantly boosting the uveoscleral outflow. NSAIDs can simultaneously reduce PGF2 and PGE2 production, while prostaglandin analogs (PGAs) have the opposite effect; they reduce the IOP by activating the PGF2 pathway. Therefore, when paired with PGAs, NSAIDs can better regulate the IOP in glaucoma patients. Tofflemire et al., 2017, discovered that diclofenac and latanoprost could lower the IOP and ease the discomfort of latanoprost in healthy horses. Additionally, it was noted that patients with primary open-angle glaucoma (POAG) may see a decrease in the IOP while taking diclofenac and latanoprost together [13]. Ketorolac can considerably increase the latanoprost, travoprost, and bimatoprost IOP-lowering effects in glaucoma patients. It is possible to utilize PGAs and ketorolac as new glaucoma treatments because they have a higher initial impact on the IOP at the start of the treatment. Additionally, Ozyol discovered that in individuals with POAG, nepafenac increases the IOP-lowering effects of various PGAs. However, after stopping the use of nepafenac, all of the study participants’ IOP levels significantly increased [14].

## 4. Treatments Targeting Proliferation: The Off-Label Approach

5-fluorouracil (5-FU) and mitomycin C (MMC) are the most used antimetabolite adjuncts to trabeculectomy surgery, and they are used as off-label medicines to limit conjunctival scarring. The use of previously approved therapeutics for off-label indications is widespread in ophthalmology. The most common drivers of off-label use are related to the marketing authorization process (e.g., long development times and high costs, limited incentives for investigating new indications), post-marketing authorization events (e.g., withdrawal from the market/product not available), pricing and reimbursement, and patient-related factors.

Access to treatments and the fulfillment of patients’ medical needs, especially in cases where no other option is available, are the primary advantages of off-label use [15].

The off-label use of medicines does not have a unique definition and regulation worldwide. The World Health Organization (WHO) [16]. defines off-label use as a sub-group of unproven interventions (i.e., an intervention for which there is insufficient evidence of safety and/or efficacy for regular use in a health system), and more specifically, an unproven mode of use of a proven intervention. Alternative definitions of off-label use are the “use of licensed medicines for indications that a national drug regulatory authority has not approved” and the “use of a pharmaceutical agent for an unapproved indication or in an unapproved age group, different dosage, duration or route of administration”. Under the WHO [17], off-label use falls under the purview of national regulatory agencies (i.e., a country-specific activity and responsibility). Therefore, every part of the world autonomously regulates the access to off-label medication use.

This picture reflects the position within the European Union, where the Member States’ approach to the off-label use of drugs is not harmonized. In some countries, special provisions about off-label use are included in the national law. In contrast, other countries have good practice guidelines/professional recommendations, are guided by reimbursement decisions, or have no policy tools.

It should be noted that authorized pharmaceuticals from one state may sometimes differ to another state in their therapeutic indication, dosage, frequency of administration, duration, or route of administration, or sometimes not be licensed at all. Consequently, it is possible that the use of a medicine licensed in State A is off-label in State B, or if used differently from the drug’s license, is also off-label in State A.

This is true for MMC. Topical MMC (under the brand name Mitosol^®^) was approved by the FDA in the United States of America in 2012 as an adjunct to ab externo glaucoma surgery. It is intended for topical application to the surgical site of glaucoma filtration surgery through sponges. More recently, topical MMC was approved by the Ministry of Health, Labour, and Welfare as an adjunct to open glaucoma surgery in Japan. However, in other countries, such as Italy, the use of MMC in ophthalmic procedures (i.e., adjuvant in anti-glaucomatous filtering surgery) remains off-label.

As for the use of 5-FU in ophthalmic procedures, from our current knowledge, it does not seem to be authorized anywhere. Therefore, its use, where permitted, is off-label.

The main issue with 5-FU and MMC treatments is their small therapeutic window, leading to unpredictable results. In some individuals, the application of the antimetabolites may not be sufficient to inhibit the fibrotic process and may result in the need for post-operative manipulations (needling of the bleb with or without injection of antimetabolites); in other individuals, they may lead to an excessive inhibition of fibrosis, which can determine hypotony, bleb leakage, blebitis, endophthalmitis, and, ultimately, vision loss. After surgery, several clinical methods were developed to classify the severity of the scarring process and to guide post-operative interventions. Recently, laser scanning of the bleb was proposed to guide the use of antimetabolites.

Another treatment used to inhibit proliferation is beta-radiation.

### 4.1. Fluorouracil (5-FU)

5-FU is a prodrug that needs to be transformed into its active metabolite 5-fluoro-deoxyuridine-monophosphate (5-FdUMP), which has an antiproliferative activity derived from the antimetabolite action in the production of pyrimidine nucleotides [18]. The presence of fluorine interferes with the conversion of deoxyuridine to thymidylic acid, depriving cells of an essential precursor for nucleic acid synthesis. 5-FU can be administered in two different ways; during the surgery, the administration can be performed with a direct sponge application, which is the most used technique, or with intraoperative subconjunctival injections. In the period after the surgery, multiple subconjunctival injections can be administered when bleb scarring is observed. The side effects of 5-FU cannot be underestimated. At high concentrations, 5-FU is toxic to all cells and can lead to increased rates of wound leakage, hypotony, and epitheliopathy compared with trabeculectomy alone [19].

### 4.2. Mitomycin C (MMC)

Mitomycin C (MMC) is an antibiotic that is isolated from *Streptomyces caespitosus,* is an alkylating agent, and acts like a DNA crosslinker. It prevents DNA synthesis, thus, efficiently modulating scarring after surgery. MMC is usually applied during surgery under the conjunctiva and/or to the scleral flap using a soaked cellulose sponge. This method of drug administration has less reproducibility than the injection route, which can be used intraoperatively in the subconjunctival and subtenon spaces. Many studies compared the efficacy and safety of trabeculectomy (considered the surgical gold standard) with and without the addition of MMC.

In POAG, low-dose MMC combined with more aggressive postoperative care improves the trabeculectomy’s results with a low rate of sequelae [20]. The analysis of all these studies shows how using MMC-implemented trabeculectomy became the standard of care.

At the same time, MMC has many potential limitations. but it is the most used antimetabolite. Furthermore, developing new agents that control postoperative scar tissue formation without side effects would be valuable and justified.

## 5. Beta-Radiation

Beta-radiation can counteract cell growth and act as an antifibrotic agent. A meta-analysis including four studies showed that intraoperative beta irradiation reduces the risk of surgical failure compared to trabeculectomy alone. Several studies demonstrated that between beta-radiation and intraoperative 5-FU, there was no difference in the risk of complications following phaco-trabeculectomy. A recent study shows that the combination of beta-radiation with MMC is superior to MMC alone. However, the beta-radiation group had a higher incidence of post-operative cataract formation [21]. The clinical use of beta-radiation is mainly limited by practical reasons, including the initial capital outlay and the limitations imposed by most countries asking for a governmental license to use radioactive devices, and the need for a dedicated lead-lined storage repository.

## 6. Monoclonal Antibodies

Biological drugs were investigated due to their anti-scarring potential. The monoclonal antibodies that are most active against growth factors and cytokines were extensively studied. We report the results obtained with bevacizumab and ranibizumab, which bind VEGF, and Infliximab, which binds tumor necrosis factor (TNF-alpha).

### 6.1. Bevacizumab

Bevacizumab is a recombinant humanized anti-VEGF immunoglobulin G1 (IgG1) approved as an antiangiogenic agent for treating metastatic colorectal cancer in combination with chemotherapy [22]. It can downregulate the mitogenic, angiogenic, and permeability-enhancing effects of VEGF-A by binding its receptors VEGFR-1 and VEGFR-2. As a potent inducer of angiogenesis, VEGF also promotes the early migration of inflammatory cells and fibroblasts. Studies on various isoforms of VEGF show that VEGF165 and VEGF121 predominantly affect blood vessel growth, whereas VEGF189 is more critical to fibrosis [23]. Bevacizumab, injected intravitreally, is used off-label to treat the neovascular macular degeneration and vascular diseases of the retina characterized by an edematous and exudative component, such as diabetic retinopathy and central retinal vein occlusion. According to different studies, the intravitreal dosage is between 1.0 and 2.5 mg; these drug quantities are administrated monthly and with varying administration regimes. The side effects of the intra-vitreal administration are less critical than the ones of the systemic administration.

In 2022, a study was conducted to determine whether subconjunctival bevacizumab or 5-FU was more effective in glaucoma patients after trabeculectomy. Fifty-one patients’ eyes, selected initially to undergo MMC trabeculectomy, were divided into three groups. A total of 13 patients were randomized to a placebo control group receiving a saline injection, 21 received 0.1 mL of 50 mg/mL 5-FU, and 17 received 1.25 mg (0.05 mL) of bevacizumab. After surgery, the IOP and bleb features were monitored one day, one week, three weeks, six weeks, six months, and one year post-operatively. Between the baseline visit and the last 1-year post-operative appointment, the IOP reduction was significantly lower than the pre-operative reduction (*p* < 0.05) in all three groups. The final average IOP between the baseline visit and the one-year follow-up were not significantly different. After a year of follow-up, the bleb evaluation was conducted utilizing the Moorfields Bleb Grading System (MBGS) [24]. The bevacizumab group had a central bleb area that was significantly wider than the 5-FU group. The bevacizumab group demonstrated a significantly lower vascularity in the central bleb than the other two groups.

Van Bergen et al. [25] also demonstrated in a 12-month, prospective, randomized, double-masked, placebo-controlled trial that a pre-operative administration of intracameral bevacizumab (1.25 mg) significantly reduces the need for additional interventions during the follow-up of patients undergoing trabeculectomy, compared to the placebo (balanced salt solution).

### 6.2. Ranibizumab

Ranibizumab is a recombinant humanized monoclonal antibody fragment (Fab) directed against extracellular endothelial VEGF. VEGF is a protein that is essential for biological processes, initiating both the physiological and pathological development of new blood vessels, and regulating vessel permeability and inflammation, all of which are believed to contribute to the development and progression of macular degeneration. This molecule is approved for on-label use in the treatment of several sight-threatening retinal disorders. Emerging evidence demonstrated that anti-VEGF treatment can offer advantages in managing other ocular conditions where VEGFs play a crucial role, such as ocular scarring following glaucoma filtering surgery and neovascular glaucoma (NVG) [26]. The evidence concerning the use of ranibizumab in glaucoma surgery is more limited compared to bevacizumab because of its higher cost. Recently, Liu et al. [26] compared the efficacy and safety of trabeculectomy (with intravitreal ranibizumab injection) to the Ahmed valve implantation. A total of 37 eyes of 36 patients with neovascular glaucoma were included in their prospective study. Eighteen eyes underwent ranibizumab injection into the vitreous body seven days before the trabeculectomy. Nineteen eyes underwent the Ahmed glaucoma valve (AGV) implantation. Patients with combined therapy (intravitreal ranibizumab injection combined trabeculectomy) showed a better outcome, a significant IOP reduction, an improvement in the best-corrected visual acuity, and a lower complication rate. 

### 6.3. Infliximab

Infliximab is a chimeric human–murine IgG1 monoclonal antibody produced in murine hybridoma cells via recombinant DNA technology. It acts specifically against human TNF-α and is traditionally administered in patients with autoimmune diseases, such as Crohn’s disease or rheumatoid arthritis [27]. A randomized study on 30 albino rabbits investigated the use of Infliximab as an adjunct in filtering glaucoma surgery. The animals were divided into the following three groups: (A) animals that received an intraoperative injection of Infliximab with different doses (from 1.0 to 5.0 mg in 0.1 mL); (B) animals that received MMC at a concentration of 0.2 mg/mL; and (C) the placebo group that received a balanced salt solution (BSS) injection. Animals were sacrificed fifteen or thirty days after surgery, and the degree of scarring was analyzed histologically. In the group of animals treated with Infliximab, the scarring level was higher, the vascularity was increased, and the bleb survival was lower than in the other two groups, suggesting that the Infliximab doses used in this study resulted in a subconjunctival TNF-α, which acted as stimulation to the fibroblasts [28]. Another study on New Zealand White rabbits suggests that topical Infliximab effectively suppresses the subconjunctival wound healing response after experimental glaucoma filtration surgery, reducing the MNC (inflammatory mononuclear cells) and fibroblast numbers and immunostaining intensities of TGF-β, FGF-β, and PDGF [29]. These conflicting results need further investigation to consider the safety of Infliximab compared to MMC.

## 7. Targeting TGF-β Signal Transduction

TGF-β is a multifunctional peptide that controls proliferation and cell differentiation synthesis and accelerates the transformation of fibroblasts into myofibroblasts [30]. The results of studies on monoclonal antibodies against TGF-β showed a reduction in scarring but a similar bleb survival after glaucoma surgery [31]. Growth factor inhibitors that interfere with TGF-β-mediated pathways were used as antifibrotics in other fibrosing disorders. A novel protein, the S58 aptamer, directed at TGF-β receptor II, can attenuate TGF-β2-mediated transdifferentiation of fibroblasts into myofibroblasts.

Lerdelimumab (CAT-152) is a monoclonal antibody targeting TGF-β2 that, in vitro, has shown the ability to reduce proliferation, fibroblast migration, and cell contraction. In vivo, it was safe and well tolerated when used on animal models after repeated subconjunctival administrations (1 mg/mL, respectively on days 0, 1, 2, 3, and 7 following surgery). Compared with MMC, this anti-TGF-β2 mAb was a potentially more controlled alternative as an anti-scarring agent in glaucoma surgery without apparent complications. The microscopic characteristics of anti-TGF-β2 mAb, compared with those of the MMC treatment, highlighted these differences. In an initial small randomized controlled human trial, the drug appeared to be promising as it guaranteed a better IOP control at three and six months compared to the placebo. However, a more extensive phase III study failed to show a significant difference in the success of the trabeculectomy between the groups of MMC and CAT at 12 months. In this trial, 388 patients were recruited and were analyzed by intent-to-treat (ITT) criteria. Considering the side effects, using lerdelimumab did not increase their incidence, and its immunogenicity was low. Still, there was no difference between the administration of lerdelimumab and the placebo in preventing the failure of primary trabeculectomy. The trial’s failure is probably related to an inadequate posology of the anti-TGF-β2 [32].

## 8. Decorin

Decorin is a small cellular or pericellular matrix proteoglycan and is closely related in structure to biglycan proteins. This proteoglycan is a component of connective tissue, binds to type I collagen fibrils, and plays a role in matrix assembly. Decorin antagonizes TGF-β signaling. When decorin is lacking, the activity of TGF-β on the outflow pathways is amplified with an increased IOP. Conversely, enhanced TGF-β signaling downregulates the expression of decorin [33]. The subconjunctival use of decorin was evaluated in 35 rabbits that were divided into five groups. The administration of decorin was well tolerated, with no signs of inflammatory reaction or infection. Anatomical damage, corneal edema, cataract formation, or endophthalmitis were not detected in the animals. Wound healing was evaluated clinically by assessing the loss of conjunctival transparency and thickening due to the deposition of fibrotic tissue. Although bleb was not evident, the conjunctiva remained translucent due to the suppression of scarring. This outcome was mainly observed in the groups of animals that received 100 µg of decorin via subconjunctival injection dissolved in 0.1 mL of vehicle solution 15 min before surgery and on the four post-surgery days [34], and in the animals that received 100 µg of decorin before surgery and on the first, third, and seventh days after the operation. The macromolecule applied locally at the highest dose and with several injections in the pre-operative and immediate post-operative period reduced the fibrotic events following the surgical trauma; it was also safe and well tolerated. However, further studies on the long-term effects and post-operative safety are necessary. Compared to CAT—152, this macromolecule can be more potent due to its inhibitory activity on TGF-β, PDGF, and CTGF [35].

## 9. S58 Aptamer

Aptamers are small, highly structured DNA/RNA oligonucleotides and can bind to a target protein [36]. Some essential characteristics of aptamers are their high selectivity, binding affinity to proteins, and very low toxicity [37]. Some in vitro and in vivo studies proved that the S58 aptamer could reduce TGF-β2 activity. A comparison with MMC showed that in the treatment with the S58 aptamer, a lower number of myofibroblasts were observed [38]. The metabolization of the S58 aptamer by nucleases can be prevented via binding with exosomes [39]. A study showed how the Exo-S58 (S58 aptamer coupled with an exosome) could reduce fibrosis more effectively than the naked S58 aptamer in human conjunctival fibroblast (HConF) cells and rat GFS (glaucoma filtration surgery) models [40]. In the HConF group treated with the Exo-S58, the proliferation of cells, migration, and fibrosis marker proteins notably shrunk compared to those treated with the naked S58. To the GFS rats’ group, Exo-S58 was administered via subconjunctival injections, and the difference in prolonged filtering bleb retention and reduced IOP values was evident compared to the naked S58. Several factors lessen the bare S58’s useful time. Naked, single-stranded RNA has a short circulation time and is rapidly degraded by nucleases. The recovery from a wound takes time. As a result, finding a new drug delivery method is essential for effectively administering the S58 aptamer to the conjunctiva following GFS. Nano-carrier drug delivery technologies can considerably increase the drug effectiveness and bioavailability in the eye. Dexamethasone-carrying chitosan nano micelles had good ocular tolerability and a comparatively longer retention time. The work concludes that exosomes are legitimate and secure delivery vehicles for the S58 aptamer. When delivered via exosomes, the S58 aptamer can considerably lessen cell migration, proliferation, and fibrosis in the TGF-2-induced HConFs. In the rat GFS models, the S58 aptamer delivered via exosomes had a more substantial antifibrotic impact than the S58 aptamer administered alone [37]. The in vivo rat GFS models also showed a sizable antifibrotic efficacy. The S58 aptamer is useful in avoiding scar tissue formation, but more research is needed.

## 10. Conclusions

Despite advancements in surgical technique and post-operative care, fibrosis remains a significant obstacle in reducing intraocular pressure after filtering glaucoma surgery.

Topical steroids and non-steroidal anti-inflammatory agents are commonly used to manage scarring in patients who undergo filtering surgery. Synthetically engineered macromolecules such as glucosamine and cyclodextrins were developed to enhance steroid delivery and effectiveness, but their use is limited in clinical practice. Targeting proliferation through off-label mitomycin C and 5-fluorouracil can lead to side effects but remains the gold standard of antifibrotic treatment during surgery and in the post-operative period. Bevacizumab and ranibizumab may provide safer alternatives, but the results are still limited and conflicting. Most recently, drugs targeting the cytokine receptors, such as the S58 aptamer, have shown promising results in reducing fibrotic tissue formation.

However, no pharmacological approaches have selectively targeted growth factors involved in the reparative process.

Overall, the data from the literature that are reviewed here clearly show that the pharmacological modulation of fibrosis following glaucoma surgery is challenging due to the complexity of the reparative process, which is characterized by the involvement of numerous factors, circulating cells, growth factors, and matrix components. Thus, therapeutic approaches aimed at multiple targets must be developed to increase the inhibitory potential against the crosstalk mechanisms, leading to excessive post-surgery scarring.

## Figures and Tables

**Figure 1 pharmaceuticals-16-00898-f001:**
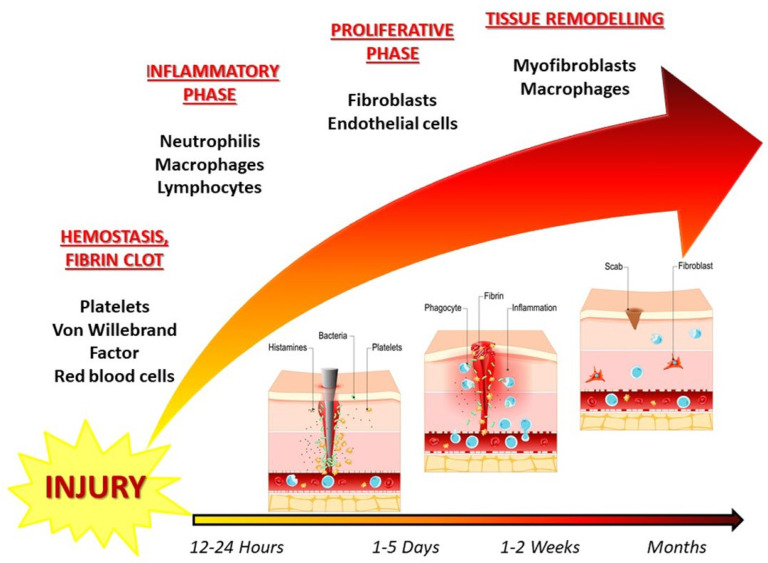
Schematic overview of the wound healing process following glaucoma surgery. The process is divided into the following four stages: hemostasis, inflammatory phase, proliferative phase, and tissue remodeling. In hemostasis, there is a platelet plug formation and the coagulation cascade activation in the first 24 h, thanks to red blood cells and platelets. In the inflammatory phase, neutrophils, macrophages, and lymphocytes are involved in the next 1–5 days. Fibroblasts and endothelial cells act in the proliferation phase one to two weeks after surgery. A scar is formed at the end of the tissue remodeling phase, thanks to myofibroblasts [6].

**Figure 2 pharmaceuticals-16-00898-f002:**
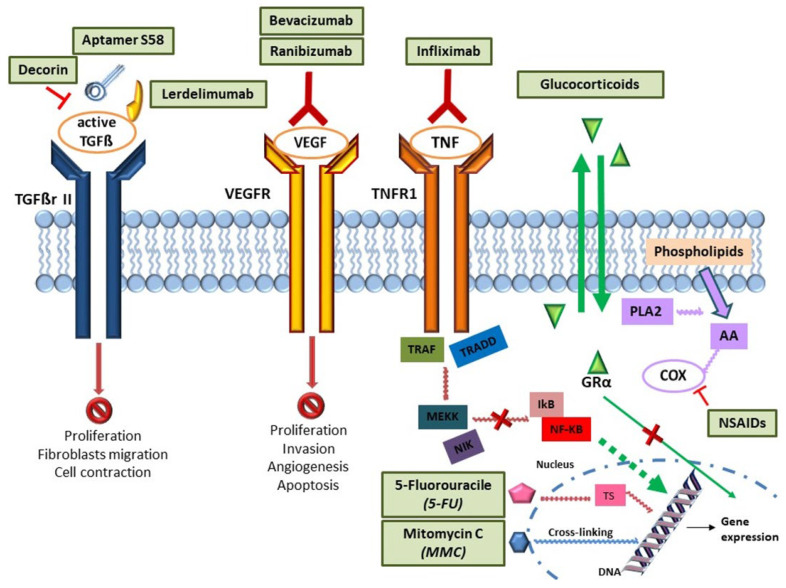
Scheme of drugs currently used and under investigation for controlling the scarring process. On the receptor, TGF beta-2 (TGFβrII), decorin, S58 aptamer, and lerdelimumab act by blocking proliferation, fibroblast migration, and cell contraction. Monoclonal antibodies such as bevacizumab and ranibizumab act on the matrix Vascular Endothelial Growth Factor (VEGF) receptor by blocking proliferation, invasion, angiogenesis, and anti-apoptosis. Infliximab works peculiarly against human Tumor necrosis factor alpha (TNF-α), avoiding the translocation of nuclear factor kappa-light-chain-enhancer of activated B cells (NF-kB) into the nucleus and its consequent gene expression, which is also inhibited by glucocorticoids. NSAIDs alleviate inflammation via inhibition of the cyclooxygenase (COX) isozymes. Finally, 5-Fluorouracile (5-FU) and Mitomycin C (MMC) are the most used antimetabolite adjuncts to trabeculectomy surgery and are used to limit conjunctival scarring. TNFR1 (Tumor necrosis factor receptor 1); PLA2 (phospholipase); AA (Arachidonic acid); NSAIDs (Non-steroidal anti-inflammatory drugs); IKB (inhibitor of nuclear factor kappa B); TS (tymidylate synthase); TRAF (TNF-Receptor Associated Factor); TNF (Receptor Associated Factors); TRADD (Tumor necrosis factor receptor type 1-associated DEATH domain protein), MEKK (stress-activated protein kinase/c-Jun N-terminal kinase); NIK (Mitogen-activated protein kinase 14 also known as NF-kappa-B-inducing kinase); GRα (glucocorticoid receptor alpha).

**Table 1 pharmaceuticals-16-00898-t001:** Drugs affecting the scarring process after glaucoma surgery.

DRUG	CATEGORY	ADVANTAGES	DISADVANTAGES
**Topical steroids**	Small molecule	Effective in controlling post-operative inflammation and edema	Increases intraocular pressure
			Cataract formation
**NSAIDS**	Small molecule	Effective in controlling post-operative inflammation and edema	Gastrointestinal side effects
		No risk of intraocular pressure spikes	Allergic reactions
**Mitomycin C**	Small molecule	Effective in preventing scar tissue formation	Bleb leaks
		Reduces fibroblast proliferation	Hypotony
**5-Fluorouracil**	Small molecule	Effective in preventing scar tissue formation	Bleb leaks
		Reduces fibroblast proliferation	Hypotony
**Bevacizumab**	Humanized anti-VEGF monoclonal antibody	Effective in reducing neovascularization and inflammation	Multiple injections
**Ranibizumab**	Humanized anti-VEGF monoclonal antibody	Effective in reducing neovascularization and inflammation	Multiple injections
**Infliximab**	Chimeric anti-TNF-alpha monoclonal antibody	Effective in reducing inflammation	Multiple injections
**Lerdelimumab**	Humanized anti-TGFβ2 monoclonal antibody	Effective in preventing scar tissue formation	Increases risk of infections
		Reduces fibroblast proliferation	
**Decorin**	Proteoglycan	Reduces accumulation of extracellular matrix and inhibits the drainage of aqueous humor	Complex injection directly into the ocular bulb
		Improves the success of trabeculectomy	
**S58 Aptamer**	TβR-II targeted oligonucleotide	Effective in reducing TGF-β2-induced fibrosis	Limits in drug ocular permeability

## Data Availability

Not applicable.

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
