# Peer review of "Pharmacological Approaches to Modulate the Scarring Process after Glaucoma Surgery"

_pharmaceuticals, 2023, doi:10.3390/ph16060898_

Round 1

Reviewer 1 Report

SUMMARY:

This manuscript by Collotta et al. is a review of both currently available and potential pharmacological approaches for modulation of scarring post-filtration surgery for the treatment of glaucoma.  The authors report that current treatments most commonly used are: topical steroids, non-steroidal anti-inflammatory drugs (NSAIDS), mitomycin C (MMC), and 5-fluorouracil (5-FU).  New approaches currently under investigation are also considered.

GENERAL:

This article will prove to be a useful resource for researchers with interest in this field.  Overall the piece is fairly well written.  However, in places, the English, although quite good, is not up to publication quality for an English-language journal.  There are also some areas where statements are unclear and difficult to follow.  I strongly urge the authors to have this manuscript reviewed by a native English speaker, or a professional English-language writing service.  Also, I would ask the authors to consider the specific points below:

ABSTRACT:

Line 17:  Control of intraocular pressure (IOP) aims to prevent further disease progression in glaucoma, but does not always succeed in doing so.  This should be stated clearly here.

INTRODUCTION:

Line 39:  Reduced aqueous humour outflow should really say, reduced aqueous humour outflow facility (or alternatively, increased aqueous humour outflow resistance).  This is a more specific statement and is in line with what has actually been measured in numerous studies focused on this parameter.  Similarly, increased production of aqueous humour would be more precisely stated as increased rate of production of aqueous humour.   

Line 45:  Please reconstruct final sentence to state something like, “Traditionally, a value of >21 mmHg (>95% CI of the mean IOP of the healthy population) is considered elevated, or ocular hypertensive.”  The way the sentence currently stands does not make sense.  An IOP value of 21mmHg is not considered “normal IOP”.

Therapeutic Approaches to Glaucoma.  Line 50.  Need to mention prostaglandin (PG) analogues here, too.    

Line 51.  The term, open-angle glaucoma, is used for the first time here.  Please briefly define.

Line 55.  The sentence here is a little muddled, and difficult for a reader less familiar with the subject matter to follow.  Please consider reconstructing as something like, “Selective laser trabeculoplasty (SLT), using a pulsed ND-Yag laser, is the most common approach for this procedure.”  Please state too that its effects tend to fade over 2 to 5 years (Line 54).  

Next paragraph (Lines 58-69).  Please briefly summarize each of the surgeries discussed.  A series of diagrams would be very helpful to the reader here.  The authors need to explain what is meant by the terms, ab externo, and ab interno.  Please emphasize again that these surgeries aim to decrease specifically, aqueous humour outflow resistance.  What is meant by the statement (Lines 63-65) that, “The surgical procedures that improve aqueous humour drainage are the first choice because they do not cause permanent damage to the anatomical structures?”  What is meant by the statement (Lines 65-66), “Surgeries improving outflow can also be classified based on which structure aqueous humour is directed”?

Also, please note that outflow surgeries these days are much more common that inflow surgeries.  I suggest then that outflow surgeries are discussed before discussing inflow surgeries.   

Figure 1.  The words, FIBRINE COLT should say FIBRIN CLOT.  Also, please include a reference to Von Willebrand Factor in this diagram as it associated with platelets.

Topical Steroids and NSAIDS:

Line 127: What is mean by the statement, “broad-spectrum anti-inflammatory drugs are commonly used as topical steroids”?  Also, please be careful to note that steroids can elevate IOP but only in steroid responder individuals. 

Line 146: What do the authors mean by the use of the word “leaned”?

Line 150:  The word “They” refers to what?    

Line 154:  PGE2 does initially cause an increase in IOP in rabbits, but later leads to a decrease in IOP.  The Blood-Ocular Barrier mentioned here is specifically known as the Blood-Aqueous Barrier.  It is important to specify as there are 2 blood-ocular barriers, namely, the Blood-Aqueous Barrier and the Blood-Retinal Barrier.

Lines 157-158:  What is meant by the statement, “prostaglandin analogs (PGAs) have the opposite effect by elevating IOP by activating the PGF2 pathway.”?  PGF analogs will reduce IOP.

Line 159:  Please add date to Tofflemire reference (2017), or else reference using the correct text citation format (13).

5-Fluorouracil (5-FU).  Line 238.  Thymidylic acid, not thymidylate acid.  Thymidylate is the conjugate base of thymidylic acid.

Beta Radiation (Lines 278-285):  Is this predominantly beta (-) radiation, beta (+) radiation, or a mix of the two?    

Ranibizumab (Line 342):  Please clarify what is mean by the word, “combined therapy”.  Is this a reference to the surgery (Trabeculectomy or AGV placement) combined with ranibizumab treatment?

Infliximab (Line 346):  What is meant by the word, “peculiarly”?  This seems inappropriate here.

TGF-B Targeting Signal Transduction (Line 368):  The phrase, “such as decorin,” is redundant here as decorin as well as CAT-152 are discussed in separated paragraphs immediately below.

References #4 and #5.  Are these two citations truly complete?  What is StatPearls [Internet]?

Overall the piece is fairly well written.  However, in places, the English, although quite good, is not up to publication quality for an English-language journal.  There are also some areas where statements are unclear and difficult to follow.  I strongly urge the authors to have this manuscript reviewed by a native English speaker, or a professional English-language writing service.

Reviewer 2 Report

The Collotta et al present a review article that explore the available and potential pharmacological treatments able to modulate the scarring process after glaucoma surgery. The study was well-written and the liteature review was adequate. In addition, the clinical revelance of this study was sound. However, I think the conclusion section in the manuscript can be cut to about half of length for refining the section.

The quality of English writing is acceptable.

Reviewer 3 Report

The paper entitled “Pharmacological approaches to modulate the scarring process after glaucoma surgery” is a review based on the current and potential pharmacological agents used to limit scarring after glaucoma surgery.

The review is quite interesting and of clinical use. Proper bleb formation and low inflammation during healing are key elements for successful long-term surgical outcomes in glaucoma. The review also provides a description of the importance, mechanisms, and uses of each agent that can be considered in surgery. A description of the new pharmacological agents currently being studied offers clinically important information for prospective studies in humans.

The review is thorough and highlights the important issues behind inflammation and underlying pathways addressed by each agent discussed. The study adds to the literature. The use of headings and subheadings gives the paper structure and a logical organization of specific correlated topics.

My compliments to the figures, which are very descriptive, useful, and nicely prepared graphically.    

Minor editing by a native English doctor can improve the English and flow of the text.

The study has been correctly planned and represents a solid basis for future studies regarding potential novel targets for treatment. It is nicely written and of clinical interest. References are appropriate. The figures and table are pertinent, and descriptive and assist in describing the results. The authors should consider a table that summarizes all drugs, with mechanisms of action, recommended dose, indications, advantages, limitations, references, etc.  

Editing by a native English doctor is suggested.
